# Legendre-SNN on Loihi-2: Evaluation and Insights

**Ramashish Gaurav**[1]*    **Terrence C. Stewart**[2]    **Yang Yi**[1]

[1]Department of Electrical and Computer Engineering, Virginia Tech, Blacksburg, VA 24060, USA
[2]National Research Council Canada, University of Waterloo, Waterloo, ON N2L3G1, Canada

## Abstract

A majority of the works on Spiking Neural Networks (SNNs) do *not* deploy and evaluate their network on a neuromorphic hardware. This not only limits the credibility of their claims of energy-efficiency and latency gains, but also discounts the opportunity to appraise neuromorphic technology for real-world computing. Herein, we especially study the technical facets of *deploying* and *evaluating* a recently formulated *State-Space Model* based spiking network called "Legendre-SNN" on Loihi-2 neuromorphic hardware. Legendre-SNN is a highly resource-efficient *reservoir*-based univariate Time-Series Classification (TSC) model. This work's emphasis is not only on its deployment on Loihi-2, but also on leveraging the Loihi-2 embedded Lakemont (LMT) cores for its *non-spiking* reservoir deployment and spike encoding. Since the documentation to program LMT is very limited, researchers often implement their *non-spiking* operations on *less* power-efficient CPUs (than LMTs). Here, we present the technical know-how to program LMT (as part of our reservoir deployment) that can be employed by later works. In our evaluation of Legendre-SNN on Loihi-2 hardware, we pleasantly find that it outperforms a complex LSTM-Conv integrated architecture on 3 of 15 datasets. We also present the *energy* & *latency* metrics of Legendre-SNN on Loihi-2, where we find that (for our settings) the reservoir on LMT consumes more than $85\%$ of total energy; consequently, we advocate for a *spiking* reservoir in Legendre-SNN.

## 1 Introduction

AI computing with Spiking Neural Networks (SNNs) is progressively gaining traction in the wake of calls for Green AI [1, 2, 3]. Training Deep Learning (DL) models on power-hungry GPUs is highly energy intensive [4, 5], and inference too - consumes notable amount of energy, e.g., ResNet-$50$ consumes more than 2.5K Joules per sample in inference mode [6]. SNNs on the other hand have proven highly energy efficient when deployed (generally for inference) on neuromorphic hardware [7], e.g., Loihi [8, 9]. Note that SNNs are still trained primarily on GPUs with methods falling under two categories: *ANN-to-SNN conversion* and *Direct Training*. Under *ANN-to-SNN conversion*, an ANN is first trained via conventional Back-propagation followed by its *conversion* to an isomorphic SNN [10, 11, 12]. Under *Direct Training*, an SNN is *directly* trained either via non-gradient methods [13, 14] or by accounting for its non-differentiable spike function [15], e.g., SLAYER [16].

Time-Series Classification (TSC) is one popular AI domain that has ubiquitous applications in Edge Computing/Smart Wearables/IoT devices. RNNs and their specialized types, e.g., GRUs and LSTMs have proven quite effective for a variety of TSC tasks, however, at the cost of *high* energy consumption - owing to their deployment on multi-CPUs/GPUs. Training RNNs is also computationally complex and time consuming in contrast to training Reservoir Computing (RC) networks [17], where, the recurrent *reservoir* is kept static, and learning adapts only the *linear readout* layer. RC networks have outperformed RNNs on a variety of tasks with ample room for improvements [17]. Note that the simplicity of RC networks also allows for their efficient hardware implementations [18]. A few popular RC models are Echo State Networks (ESNs) [19] and Liquid State Machines (LSMs) [20], where the ESNs are composed of *non-spiking* neurons and LSMs are composed of *spiking* neurons.

---

*Corresponding author: rgaurav@vt.edu, https://r-gaurav.github.io/

38th Second Workshop on Machine Learning with New Compute Paradigms at NeurIPS 2024(MLNCP 2024).

Recently, a novel reservoir-based SNN called *Legendre-SNN* (LSNN) was formulated by Gaurav et al. [21] that outperformed the LSM-based models. The LSNN consists of a static reservoir followed by an encoding layer and a trainable *non-linear* spiking readout layer – unlike conventional RC. The reservoir in LSNN is actually a *State-Space Model* (SSM) implemented via simple matrix operations (*no* neurons) and its output is encoded to spikes, while the non-linear readout layer comprises *one* spiking hidden layer followed by a classification layer – more architectural details in Sec 2.4. Note: the authors [21] in their ablation study found that the LSNN *without* a hidden layer still outperformed the LSM-based models; they also energy-profiled LSNN on Loihi-1 (only the post-reservoir spiking part). We chose LSNN for our deployment because it offers a *unique* blend of crucial *non-spiking &* *spiking* components that enable the utilization of Loihi-2's *two* computational resources: Lakemont cores & Neuro-Cores; also, quantized SSMs are gaining traction [22]. SSMs are dynamical systems that accept an input $u(t)$ and maintain an internal state $x(t)$, where $x(t)$ captures the latent dynamics governing the response $y(t)$; state $x(t)$ evolves over time guided by a defined *state-transition* rule, and is used to predict the future states and generate the output $y(t)$ via an *observation* rule [23]. The SSM used in LSNN is called *Legendre Delay Network* (LDN) [24, 25] – more details in Sec 2.3.

In this study, given that the authors of LSNN did *not evaluate* it (on any dataset) on a neuromorphic chip, we put its performance to test on Loihi-2. Note: the mix of *non-spiking & spiking* also makes LSNN *non-trivial* to port *entirely* on Loihi-2. With these motivations, our contributions are:

- We deploy & evaluate LSNN on *physical* Loihi-2 and report energy & latency metrics. This work can be considered one of the firsts to deploy a RC-based TSC model on *Loihi-2* [26].
- In a first (to our best knowledge), we implement a quantized SSM (i.e., the LDN) on the embedded Lakemont cores of Loihi-2 and energy-profile it. As part of this implementation:
    - We add to the *scarce* technical documentation to program Lakemont cores (Appx F)

## 2 Background

We now present the necessary background relevant to our work. We start with a short description of Loihi-2 & Lava, followed by the description of the LDN (an SSM used as reservoir) and the LSNN.

### 2.1 Loihi-2 chip

Loihi-2 [27] is the 2nd generation neuromorphic chip from Intel, which improves upon Loihi-1 [28]. Notably, Loihi-2 comes with $1,048,576$ spiking units, 120 million synapses, and 6 embedded x86 Lakemont (LMT) cores per chip; it has 128 Neuromorphic Cores (Neuro-Cores), each housing 8192 spiking units. The LMT cores are responsible for many essential tasks, e.g., facilitating spike-based communication, data I/O (i.e., encoding and decoding), and network management/configuration, etc. The embedded LMTs execute standard `C` code, however, with support for *only* 32-bit operations; this implies that *only quantized operations* can be executed on LMTs. The spiking units on Neuro-Cores are asynchronous programmable CUrrent BAsed (CUBA) neurons that generate binary (as well as, 32-bit graded) spikes; the Neuro-Cores also support Three-Factor Rules [29] based On-chip training.

### 2.2 Lava software suite

The Loihi-2 chips can be programmed using Intel's "Lava" software suite; the suite consists of multiple libraries e.g., `lava`, `lava-dl`, etc., that can be used to build and train a variety of spiking networks, along with support for their deployment and profiling on Loihi-2. A concise documentation and programming features of Lava[2] can be found in [30]; herein, we describe only those programming paradigms of Lava that are most relevant to our work. The two important `lava` paradigms used to build (our) Loihi-2 deployable networks are `Process` and `ProcessModel`. Note that a `Process` can have multiple *corresponding* `ProcessModels` depending on the hardware backend the `Process` is intended to be deployed upon. A `Process` defines the *interface* of a component of the network, while the corresponding `ProcessModel` defines the *implementation* of its `Process` (on the desired backend). Lava supports 3 types of `ProcessModels`: `PyLoihiProcessModel` (meant for `Python` implementations to be deployed on CPU), `CLoihiProcessModel` (meant for `C` implementations to be deployed on embedded LMT), and `NcProcessModel` (meant for `NxCore` implementations to be deployed on Neuro-Cores). The `ProcessModels` of different `Processes` synchronize among themselves via a *protocol* called `LoihiProtocol` that consists of multiple execution *phases* implemented by the participating `Processes` (details in Appx A). Coming to `lava-dl`, it assists in building and directly training the SNNs on GPUs (via its `slayer` [16] API), and their deployment on Loihi-2 (via its `netx` API). Lava also provides a *simulation* backend for the *physical* Loihi-2 chip; the switch to *simulation* and *physical* backends can be made via Lava's `Loihi2SimCfg` and `Loihi2HwCfg` APIs.

---

[2] `https://lava-nc.org/`

## 2.3 Legendre Delay Network (LDN)

LDN – a type of an SSM introduced by Voelker et al. [24, 25] is a neural approximation of a Linear Time-Invariant system implementing a delay of $\theta$ secs, i.e., for a univariate input signal $u(t)$ to the LDN, its output is $y(t) = u(t-\theta)$. Following are the state equations of the LDN in *continuous*-time i.e., Eqs (1) & (2), and *discrete*-time i.e., Eqs. (3) & (4) (note: Legendre-SNN uses only Eq (3)):

$$\dot{x}(t) = Ax(t) + Bu(t) \quad (1) \qquad \mathbf{x}[t+1] = \mathbf{A}\mathbf{x}[t] + \mathbf{B}\mathbf{u}[t] \quad (3)$$
$$y(t) = Cx(t) + Du(t) \quad (2) \qquad \mathbf{y}[t] = \mathbf{C}\mathbf{x}[t] + \mathbf{D}\mathbf{u}[t] \quad (4)$$

where the Eq (1) (& Eq (3)) is the *state-transition* equation and the Eq (2) (& Eq (4)) is the *observation* equation. Note that $u(t)\in\mathbb{R}$, $x(t)\in\mathbb{R}^d$, and $y(t)\in\mathbb{R}$ ($\mathbf{u}[t]$, $\mathbf{x}[t]$, and $\mathbf{y}[t]$) are the LDN's input, state-vector, and output in continuous-time (and discrete-time). The values of the state-matrices $A$, $B$, $C$, & $D$ (and $\mathbf{A}$, $\mathbf{B}$, $\mathbf{C}$, & $\mathbf{D}$) in the equations above that define an LDN are stated in Appx B.

## 2.4 Legendre-SNN (LSNN) [21]

LSNN is a highly resource efficient reservoir-based SNN for TSC tasks. In their experiments, the authors [21] used a maximum of 120 "Integrate & Fire" (IF) neurons and achieved SoTA *spiking* performance on 5 univariate-TSC datasets. The LSNN constitutes of an LDN functioning as a *static* reservoir followed by an SNN comprising an encoding layer, *one* hidden and an output layer – both trainable. The LDN accepts a univariate signal $\mathbf{u}[t]$ and outputs a $d$-dimensional state-vector $\mathbf{x}[t]$ (authors [21] refer $\mathbf{x}[t]$ as the *temporal features* of input $\mathbf{u}[t]$). The features $\mathbf{x}[t]$ are rate-encoded to binary spikes via a two-neuron (IF) encoder layer (ENC). Note that the scalars in $\mathbf{x}[t]$ can either be *positive* or *negative*, therefore, one needs a *pair* of neurons, each sensitive to the sign of the scalar, to encode. Thus, the number of neurons in ENC is $2\times d$ (henceforth, $2d$). The ENC layer is followed by a hidden layer (HDN) of $3\times d$ (henceforth, $3d$) number of IF neurons; the spikes of which are forwarded to the *non-spiking* output

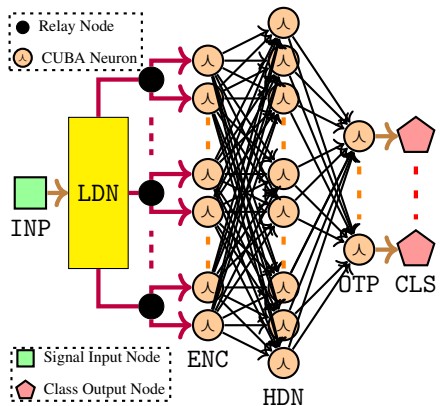

Figure 1: Our *adapted* LSNN model [21]. Connections ENC to HDN to OTP are all-to-all.

layer (OTP), where, the classes are inferred on the maximally accumulated voltage of OTP nodes. Thus, the total number of IF neurons in LSNN is $2d+3d=5d$ and the number of *trainable* connections is $2d\times 3d+3d\times c=6d^2+3dc$ (where $c$ is number of OTP nodes/classes). Compared to the original LSNN, there are *two* distinctions in our adapted LSNN (see Fig 1): (1) Instead of a *non-spiking* OTP layer, for programming constraints/ease on Loihi-2 we use a *spiking* OTP layer; we forward these OTP spikes to a new classification layer (CLS) that infers class on the maximally firing OTP neuron, & (2) Except the ENC layer IF neurons, all the neurons in HDN and OTP are CUBA neurons (details later). Henceforth, term "LSNN" implies our adapted LSNN. In Fig 1, the number of "Relay" nodes is equal to $d$, and each of them *duplicates* the corresponding scalar in $\mathbf{x}[t]$ and forwards it to ENC. The total number of neurons in our adapted LSNN is $5d+c$ and the number of trainable connections remain same (similar to [21] - only the black connections in Fig 1 are trainable). In the original LSNN, the authors considered the maximum value of $d=24$ and $c=2$; in our experiments, this implies that the number of neurons$=5\times 24+2=122$ and number of connections$=6\times 24(24+1)=3600$.

## 3 LSNN's Training on GPU, and Evaluation & Profiling on Loihi-2

We now explain our framework to train our adapted LSNN (Fig 1) on GPU and evaluate & profile it on Loihi-2. Note that the reservoir LDN in LSNN is implemented via regular matrix operations (i.e., Eq (3)) and *not* with spiking neurons; whereas, the spiking network following the LDN is composed of IF (in ENC) and CUBA neurons (in HDN & OTP). Although the LSNN as a whole can be easily trained on CPU/GPU, its evaluation & profiling on Loihi-2 is *not* straightforward, the *non-spiking* LDN and *spiking* network must be accounted appropriately; we do that by deploying the LDN along with ENC layer on the embedded LMT cores and the following HDN & OTP layers on the Neuro-Cores. However, implementing the LDN on LMT is *not* trivial either – because the LMT cores support only 32-bit signed integer operations. Therefore, to implement the LDN (& ENC layer) on LMT, we must *quantize* all its real-valued variables and operations to 32-bit integers. Note that for the uniformity of our experiments, we train the LSNN with *quantized* LDN (instead of a *continuous*-valued one). The

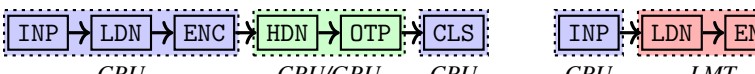



| CPU | CPU/GPU | CPU |
|:---:|:---:|:---:|

(a) Backends for training LSNN on CPU/GPU

| CPU | LMT | Neuro-Core | CPU |
|:---:|:---:|:---:|:---:|

(b) Backends for evaluating LSNN on Loihi-2



Figure 2: Training and Evaluation backends of Legendre-SNN (LSNN). Note that the backends in Fig 2b are for LSNN's deployment on *physical* Loihi-2; for its deployment on Loihi-2's *simulation*, all backends are CPU.

Fig 2 above pictorially shows our *training* and *evaluation* backends of the LSNN. We next explain our method to quantize the LDN, followed by the details of LSNN's training and evaluation.

### 3.1 Quantization of LDN

Quantization of a floating-point number is commonly achieved by *scaling* it with a power of 2 (say $\mathcal{P}$) and taking only the integer part (i.e., clipping away the fractional part after scaling). Note that the magnitude of scaling factor $\mathcal{P}$ determines the precision of quantization; the higher it is, the more information is preserved during quantization (and precision loss is low). However, considering the subsequent quantized operations and for numerical stability, $\mathcal{P}$ should be appropriately chosen so as not to overflow the bit limitation. Henceforth, we denote all the quantized variables with a bar over it e.g., the continuous-valued $\mathbf{u}[t]$ is quantized as $\bar{\mathbf{u}}[t]$, where $\bar{\mathbf{u}}[t]=\langle\mathcal{P}\times\mathbf{u}[t]\rangle$ ($\langle.\rangle$ denotes rounding to nearest integer). We next explain our quantized operations to compute the LDN's state-vector $\bar{\mathbf{x}}[t]$. We start with the quantization of Eq (3), where the matrices $\mathbf{A}$ & $\mathbf{B}$, states $\mathbf{x}[t]$ & $\mathbf{x}[t+1]$, and input $\mathbf{u}[t]$ are all floating-point. Therefore, $\overline{\mathbf{A}}=\langle\mathcal{P}\times\mathbf{A}\rangle$, $\overline{\mathbf{B}}=\langle\mathcal{P}\times\mathbf{B}\rangle$, and $\bar{\mathbf{x}}[t]=\langle\mathcal{P}\times\mathbf{x}[t]\rangle$; this implies:

$$\bar{\bar{\mathbf{x}}}[t+1] = \overline{\mathbf{A}}\bar{\mathbf{x}}[t] + \overline{\mathbf{B}}\bar{\mathbf{u}}[t] \tag{5}$$

where $\bar{\bar{\mathbf{x}}}[t+1]$ has the contribution from the factor $\mathcal{P}$ *twice*. Therefore, we scale it back as $\bar{\mathbf{x}}[t+1] = \left\lceil \frac{\bar{\bar{\mathbf{x}}}[t+1]}{\mathcal{P}} \right\rceil$ for the next iteration, where $\lceil.\rceil$ is the ceil operation. Now that we have obtained $\bar{\mathbf{x}}[t+1]$, we rate encode it to binary spikes via the ENC neurons (equations in Appx C). Note: we conduct detailed *spike-train synchrony tests* (see Appx D) between the rate encoded $\bar{\mathbf{x}}[t]$ & $\mathbf{x}[t]$ (i.e., quantized & continuous-valued respectively) to confirm the correctness of our LDN quantization process.

### 3.2 Training LSNN on GPU

Since the LDN reservoir (& ENC) in LSNN is *static* we do not train it, while the network comprising HDN & OTP layers is *trainable*. We build and train *that* network (i.e., HDN→OTP) using the lava-dl's slayer API – henceforth, we call this CPU/GPU *trainable* network as *SlayerSNN*. As can be seen in Fig 2a, the univariate INP, quantized LDN, and the two-neuron ENC layers are implemented on CPU, such that the $\bar{\mathbf{x}}[t]$'s encoded spikes are fed to the *SlayerSNN* i.e., the HDN and OTP layers network composed of CUBA neurons; we keep the *current* & *voltage* decay hyper-parameters ($\tau_{\text{cur}}$ & $\tau_{\text{vol}}$) of these neurons tunable. We leverage the SpikeRate loss defined in slayer over the spiking rate of the OTP layer neurons and the Adam optimizer [31] to train *SlayerSNN*; the classes are inferred on the maximally firing OTP neuron. Note that in each training epoch, we also evaluate the trained LSNN on the test data and save the network parameters that obtains the best test accuracy. The saved *SlayerSNN* is then later used along with the quantized LDN to obtain a Loihi-2 deployable LSNN.

### 3.3 Evaluating LSNN on Loihi-2

A network composed of Lava Processes can be deployed on *physical* Loihi-2 chips, as well as on Loihi-2's *simulation* running on CPUs. Therefore, we build our Loihi-2 deployable LSNN from the saved best-performing *SlayerSNN* using Lava Processes; henceforth, we call this Loihi-2 deployable LSNN as *LavaLSNN*. We use lava-dl's netx API to *exchange* the saved *SlayerSNN* to a Lava Process (say *NetxSNN*) that we interface with other constituent Processes of our *LavaLSNN*. The other three main Processes of *LavaLSNN* are: *InpSigToLdn*, *LdnEncToSpk*, and *OutSpkToCls*; each of these three Processes have *one or more corresponding* ProcessModels for the backends they are intended to be deployed upon (see Appx E). Fig 3 shows the *architecture* of our *LavaLSNN* and the supported backends for the constituent Processes. Note that *InpSigToLdn* Process feeds an input signal to the LDN, *LdnEncToSpk* Process implements the quantized LDN and encodes its $\bar{\mathbf{x}}[t]$ to binary spikes, which are sent to the *NetxSNN*, and *OutSpkToCls* Process infers classes from the *NetxSNN*'s output spikes (and computes accuracy). Also note that since a core part of our contributions is the programming of LDN on LMT, we present its technical know-how in Appx F.

## 4 Experiments & Results

We now present the details of our experiments conducted with LSNN and related results. We experiment on 15 TSC datasets with number of classes $\in[2,5]$, and signal duration $\in[140,900]$ time-steps

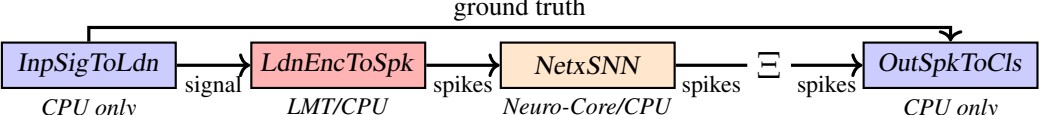

Figure 3: *LavaLSNN* `Processes`, their connections and supported backends; 'Ξ' is an adapter `Process` to adapt spikes from Neuro-Core to CPU. For *LavaLSNN*'s *physical* Loihi-2 deployment, blocks are color coded with Fig 2b (backends are: LMT & Neuro-Core); for deployment on Loihi-2 *simulation*, all backends are CPU.

(see Appx G). The *two* parameters that characterize the LDN (in LSNN) are its delay value $\theta$ and dimension $d$ of its state-vector $\mathbf{x}[t]$ (these values define the state-matrices $\mathbf{A}$ & $\mathbf{B}$). In our experiments, we tune $\theta \in \{110, 130, 150\}$ time-steps, and $d \in \{4, 6, 8, 10, 12, 14, 16, 24\}$; the CUBA neurons' $\tau_{\text{cur}}$ and $\tau_{\text{vol}}$ decays are tuned in $\{0.00, 0.10, 0.20\}$. The batch size and number of training epochs vary with dataset and can be found in our code [32]. For all the experiments we shuffle the train and test data every $20^{\text{th}}$ epoch and halve the learning rate $\eta$ every $40^{\text{th}}$ epoch (initial $\eta = 0.001$). Note that we set the factor $\mathcal{P}=4096$, such that the operations in Eq (5) are well represented in 32-bit.

### 4.1 Training and test procedure on CPU/GPU

As stated above, we train our LSNN (*SlayerSNN* specifically, with quantized LDN) on GPU (see Fig 2a). For each dataset, we conduct grid-search on the abovementioned values of $\theta$, $d$, $\tau_{\text{cur}}$, & $\tau_{\text{vol}}$ (we ensure $\theta \leq$ signal duration). Let $\kappa_m$ denote one combination of $(\theta_i, d_j, \tau_{\text{cur}_k}, \tau_{\text{vol}_l})$; for each $\kappa_m$ we train the LSNN on 5 different $s_n \in$ `SEEDs`. Note that for each of the training runs for a dataset, we obtain the best performing LSNN test-accuracy: $slyr\_acc_{s_n}^{\kappa_m}$, and use the same trained network to obtain the *corresponding LavaLSNN* accuracy: $l2sim\_acc_{s_n}^{\kappa_m}$ on Loihi-2 *simulation* i.e., on CPU. We then compute the final test accuracies reported in Tab 1 as follows (where $\mathsf{mdl}=slyr$ or $l2sim$):

$$\mathsf{mdl\_best} = \max_{\kappa_m, s_n} \left(\mathsf{mdl\_acc}_{s_n}^{\kappa_m}\right) \quad , \quad \mathsf{mdl\_mean} = \max_{\kappa_m}\left(\operatorname*{mean}_{s_n}(\mathsf{mdl\_acc}_{s_n}^{\kappa_m})\right) \quad (6)$$

### 4.2 Evaluation procedure on *physical* Loihi-2

We evaluate our *LavaLSNN* on *physical* Loihi-2 for all the 15 datasets; Fig 2b shows *LavaLSNN*'s respective backends for deployment. Since the Loihi-2 boards on Intel's INRC cloud are on shared access, we prioritize the variety of *LavaLSNN* models that we evaluate on *physical* Loihi-2. Therefore, for each dataset, we first choose and evaluate *all* those *LavaLSNN* models that correspond to $l2sim\_best$, followed by evaluating a next few models with $l2sim\_acc_{s_n}^{\kappa_m}$ within 2% or 3% absolute margin of $l2sim\_best$ accuracy; we then report the best accuracy (among all) - $l2hw\_best$ in Tab 1.

### 4.3 Results Analysis

We now analyze our results in Tab 1, where we compare ours with that of LSTM-FCN [33]. LSTM-FCN is composed of *one* LSTM block, *three* 1D Convolution blocks (each accompanied with batch normalization), and dropout & average pooling blocks. As the LSTM-FCN is a significantly complex architecture than the LSNN and the related work [33] provides a benchmark on all our experimented datasets, we choose it for comparison. In Tab 1, the simple LSNN either achieves the *same accuracy* or *outperforms* the well established LSTM-FCN on 5 datasets – considering the best and average performance of both: LSNN on GPU ($slyr\_*$) and on Loihi-2 *simulation* ($l2sim\_*$). Regarding the variation of mean($slyr\_acc_{s_n}^{\kappa_m}$) and mean($l2sim\_acc_{s_n}^{\kappa_m}$) (over `SEEDs` $s_n$) with hyperparameters $d$, $\theta$, $\tau_{\text{cur}}$, & $\tau_{\text{vol}}$, we did *not* find any conclusive trend between them (similar to [21]); although, for a few datasets, we observed that higher $d$ results in higher accuracy. Coming to LSNN's performance on *physical* Loihi-2 ($l2hw\_best$), we see it *outperforming* LSTM-FCN on 3 datasets. It is expected that a typical network's deployment on a hardware and its simulation should produce same results, however, we did *not* observe this in our Loihi-2 experiments. In fact, for a few datasets, we observed that *non* $l2sim\_best$ models achieved better accuracies on Loihi-2 hardware; here too, we observed that *generally* higher $d$ achieved better accuracies. Regarding comparison with other spiking networks, we found other *spiking* works on 8 of our experimented datasets. For the first five datasets in Tab 1 -[21] achieved (max) 98.49, 93.56, 82.72, 99.51, 80.43 (%) accuracies (resp.), for *Cofe* & *Light2*: -[34] achieved 100.0 & 80.33 (%) accuracies (resp.), and for *ToSe2*: -[35] achieved 83.00%. Note: [21] tune their LSNN on a much wider set of hyper-parameters with a different loss function, and [34] use trainable `ReLU Conv` layers to extract temporal features – hence, they obtain higher accuracies. Additionally, we conducted two *intrusive* experiments with LSNN, where (a) we used *continuous* valued LDN and (b) the neuron decays ($\tau_{\text{cur}}$ & $\tau_{\text{vol}}$, along with the weights) were set to trainable. For *ToSe2* dataset, upon inferring with such an $l2sim\_best$ model on *physical* Loihi-2, we obtained 93.08% accuracy (same as LSTM-FCN, Tab 1); we present more details in Appx H.

## 5 Discussion & Insights

We now profile the LSNN on *physical* Loihi-2 and present some insights from its analysis.

Table 1: Our results with LSNN on GPU, on Loihi-2 *simulation*, and on *physical* Loihi-2 hardware

| Method/ Accuracy | Datasets | | | | | | | | | | | | | | |
|---|---|---|---|---|---|---|---|---|---|---|---|---|---|---|---|
| | E5000 | FordA | FordB | Wafer | Eqaks | Cofe | ToSe1 | ToSe2 | Light2 | Wrms | WrTC | OlOil | Meat | Comp | RDev |
| slyr_best | 95.51 | 91.29 | 81.61 | 97.96 | 79.14 | **100.0** | 93.86 | **93.85** | 78.69 | **80.52** | **87.01** | 66.67 | **100.0** | 71.60 | 54.40 |
| slyr_mean | 94.95 | 90.86 | 79.73 | 97.70 | 76.26 | 95.71 | 91.75 | 88.92 | 72.13 | **75.32** | **84.68** | 46.68 | **95.33** | 68.56 | 48.75 |
| l2sim_best | 95.31 | 90.98 | 80.12 | 98.26 | 76.98 | **100.0** | 92.98 | **93.85** | 78.69 | **79.22** | **85.71** | 63.33 | **96.67** | 69.60 | 53.33 |
| l2sim_mean | 94.83 | 90.71 | 78.57 | 97.91 | 75.40 | 90.71 | 91.49 | 88.31 | 71.80 | **75.06** | **84.94** | 43.33 | **92.00** | 67.68 | 49.39 |
| l2hw_best | 95.33 | 90.61 | 77.90 | 98.38 | 77.70 | 92.86 | 92.11 | 90.77 | 68.85 | **75.32** | **84.42** | 46.68 | **98.33** | 70.40 | 50.40 |
| LSTM-FCN | 94.73 | **92.72** | **91.80** | **99.92** | **83.54** | **100.0** | **98.25** | 93.08 | **80.33** | 66.85 | 79.56 | **86.67** | 91.67 | **86.00** | **58.13** |

Note: We binarized E5000 dataset from 5 to 2 classes (as done in [21]); on E5000, LSTM-FCN obtains accuracy on all the 5 classes, whereas we experimented with binarized classes, therefore we do not highlight in bold. The terms *slyr_\**, *l2sim_\**, and *l2hw_\** denote accuracies on GPU (from SLAYER), on Loihi-2 *simulation* (on CPU), and on *physical* Loihi-2 respectively. The datasets' names and descriptions are in the Appx G; all numerals in this table are accuracies in %. For *\*_mean* results, $std$ for all the datasets $\leq 0.05$; for a few datasets, $std$ approached $0.001$.

## 5.1 Profiling LSNN on *physical* Loihi-2 chip

To profile SNNs on Loihi-2, it is recommended to run workloads uninterrupted for sufficiently long - in repeats. Also, when profiling with energy-probes, two *settings* that control the probe's buffer *size* ($\mathcal{S}_j$) and its information *binning* over ($\mathcal{B}_k$) time-steps need to be adjusted for sensible measurements. Note: our emphasis here is *not* on proving energy-efficiency of SNNs on Loihi-2 ([8, 9] already do), rather on profiling the *individual* components/backends of *LavaLSNN* (see Figs 2b & 3). Especially, we are interested in the energy consumption of (1) LMT & Neuro-Core i.e., *LdnEncToSpk→NetxSNN*, (2) only LMT i.e., *LdnEncToSpk*, and (3) only Neuro-Core i.e., *NetxSNN*. An important nuance to consider here is that the *LavaLSNN* has a signal input `Process` on CPU. The communication between CPU & chip introduces a lot of variability in execution time and energy consumption; this overhead also brings instability in running longer workloads. Hence, for stability and consistency, we hard-code a $140$ time-steps (t.s.) quantized sample signal from *E5000* dataset on LMT itself (in its `C` code) and conduct all our profiling; for (3), as no LDN/LMT is involved, we stimulate the trained *NetxSNN* via *biased* neurons (right on Neuro-Core) that spike every few t.s. The detailed procedure to obtain LSNN's energy and latency metrics (for three different orders $d$) reported in Tab 2 is mentioned in the Appx I; note that *two* Neuro-Cores and *one* LMT core were utilized. In Tab 2, we clearly see that (2)-*LdnEncToSpk* on LMT expectedly consumes the major chunk of energy, while (3)-*NetxSNN* on Neuro-Cores consumes minimal. In fact, for $d$=24, we see the energy metrics of (2) & (3) add up to that of (1); latency for (3) varied with $\mathcal{B}_k$ (all $< 2ms$).

The LSNN authors [21] chose a *non-spiking* LDN to improve on their precursor - Spiking Reservoir Computing (SRC) model [36]. The SRC uses Neural Engineering Framework (NEF) principles [37] to implement a *spiking* LDN (on Loihi-1); compared to it, the *non-spiking* LDN helps in improved feature *representation* and *computation*, but that incurs the cost of more than $85\%$ energy proportion (for $d$=24, between (2):(1) in Tab 2). This recommends going *back* to choosing *spiking* implementation of LDN in LSNN for reaping the most of Loihi-2's energy-efficiency (see Appx J). The *spiking* LDN may affect LSNN's performance, however, training methodology is also crucial, not just quality of features. Note that LSNN employs gradient-based optimization with a dedicated *classification* loss function (unlike SRC's *regression* one), and even without a `HDN` layer, it outperforms SRC [21]; thus, using a *spiking* LDN in LSNN is promising from performance & energy perspective.

## 6 Conclusion & Future Work

In this work we evaluated a highly *resource-* & *energy*-efficient reservoir-based TSC model on Loihi-2. We also presented the *first implementation* of a quantized SSM (i.e., LDN) on the LMT cores of Loihi-2 (& added to the *scarce* documentation to program LMTs); our *implementation* can easily be adapted to other SSMs by accordingly setting the state-matrices. Considering the perceived inferiority of SNNs, our results (in Tab 1) are noteworthy – *LavaLSNN* on *physical* Loihi-2 *outperforms* LSTM-FCN on 3 datasets and *performs at par* on another. Our LSNN's energy-profiling decisively ascertains the need of *spiking* LDN in LSNN (from energy perspective) - with optimism for similar or better performance. A *spiking* LDN will also forgo the need of `ENC` layer, although, implementing a *spiking* LDN (via NEF) on Loihi-2's Neuro-Cores is convoluted due to no dedicated Lava support.

Table 2: Per-sample Energy (Enrg) & Latency (Ltnc) metrics of *LavaLSNN* components on Loihi-2

| | (1) LMT & Neuro-Core | | | (2) only LMT | | | (3) only Neuro-Core | | |
|---|---|---|---|---|---|---|---|---|---|
| Order $d\rightarrow$ | 8 | 16 | 24 | 8 | 16 | 24 | 8 | 16 | 24 |
| **Enrg** ($mJ$) | 2.46, 0.26 | 2.45, 0.29 | 2.88, 0.29 | 2.44, 0.28 | 2.46, 0.27 | 2.52, 0.31 | 0.31, 0.09 | 0.31, 0.09 | 0.32, 0.09 |
| **Ltnc** ($ms$) | 11.46, 1.19 | 11.22, 1.15 | 13.13, 1.26 | 11.33, 1.13 | 11.46, 1.09 | 11.60, 1.33 | - | - | - |

Metrics for (1), (2): from hard-coded signal on LMT, for (3): from stimulation on Neuro-Core; $(x,y)\Rightarrow$(mean, std)

## Acknowledgment

We thank the anonymous reviewer(s) for their inputs to improve the presentation of this work, and Intel for providing us the access to Loihi-2 boards on their INRC cloud. This work was supported in part by the U.S. National Science Foundation (NSF) under Grant CCF-1750450, Grant ECCS-1731928, Grant ECCS-2128594, Grant ECCS- 2314813, and Grant CCF-1937487.

# Appendix

## A `LoihiProtocol`

Here we briefly describe the `LoihiProtocol` used to synchronize the Lava `Processes`, more details can be found at `https://lava-nc.org/`. The *phases* in `LoihiProtocol` are executed in a *particular order*, and all the phases (except one) have an implementable *guard* method that returns a boolean value - determining if its corresponding phase should be executed in a given time-step. The phases and their guard methods are described below; phases are in the order of their execution:

1. Spiking phase (`run_spk()`): This phase is executed $1^{st}$ in order and has *no* guard method, i.e., it is executed unconditionally every time-step.

2. Pre-management phase (`run_pre_mgmt()`): Executed $2^{nd}$ in order, and has a guard method `pre_guard()` that must return `True` along with `lrn_guard()` to execute this phase.

3. Learning phase (`run_lrn()`): Executed $3^{rd}$ in order, and its guard method `lrn_guard()` must return `True` for this phase to be executed.

4. Post-management phase (`run_post_mgmt()`): Executed $4^{th}$ in order, and its guard method `post_guard()` must return `True` for this phase to be executed.

5. Host phase (`run_host_mgmt()`): Executed $5^{th}$ in order, its guard method `host_guard()` must return `True` for this phase to be executed.

Note: one can choose to skip a few phases depending on their implementation requirements.

## B LDN's State-matrices

Below, we mention the values of the state-matrices $A$, $B$, $C$, and $D$ that define an LDN - characterized by the dimension $d$ (of its state-vector $x(t)$) and delay $\theta$ (i.e., its encoding memory window):

$$A = \frac{[a]_{i,j}}{\theta}, \quad a_{i,j} = (2i+1)\begin{cases} -1 & i < j \\ (-1)^{i-j+1} & i \geq j \end{cases} \qquad \text{and} \qquad B = \frac{[b]_i}{\theta}, \quad b_i = (2i+1)(-1)^i \tag{7a}$$

$$C = [c]_i, \quad c_i = (-1)^i \sum_{j=0}^{i} \binom{i}{j}\binom{i+j}{j}(-1)^j \qquad\qquad \text{and} \qquad\qquad D = 0 \tag{7b}$$

for $i, j \in [0, d-1]$. Note: $\mathbf{A}$, $\mathbf{B}$, $\mathbf{C}$, and $\mathbf{D}$ in Eqs. (3) & (4) are obtained from $A$, $B$, $C$, and $D$ via Zero-Order Hold (ZOH) method – Eq. (8) below:

$$\mathbf{A} = e^{(A\Delta t)}, \qquad \mathbf{B} = A^{-1}(e^{(A\Delta t)} - I)B, \qquad \mathbf{C} = C, \qquad \text{and} \qquad \mathbf{D} = D \tag{8}$$

where $I$ is an Identity matrix, $e^{(\cdot)}$ is *matrix exponential* function, and $\Delta t = 1ms$.

## C LSNN's neuron equations

Here we present the neuron equations of our adapted LSNN.

### C.1 `ENC` layer

We rate encode the quantized LDN state-vectors $\bar{\mathbf{x}}[t]$ to binary spikes $S[t]$ (using Heaviside $\Theta(.)$).

$$\bar{J} = \alpha < \mathcal{E}.\bar{\mathbf{x}}[t] > +\bar{\beta} \quad , \quad \bar{V}[t] = \bar{V}[t-1] + \bar{J} \quad \text{and} \quad S[t] = \Theta(\bar{V}[t] - \bar{V}_{thr}) \tag{9}$$

where $\alpha$ (=1) & $\bar{\beta}$ (=0) are the neuron's *gain* & *bias* values, $\mathcal{E}$ is encoder vector $\in \{-1, 1\}$, and $\bar{V}[t]$ & $\bar{V}_{thr}$ are the neuron's quantized *voltage* & *threshold* respectively (we set $V_{thr}=1$, $\implies \bar{V}_{thr}=\mathcal{P}\times 1$).

**C.2** `HDN` **&** `OTP` **layers**

Following are the equations of the CUBA neurons used in `HDN` and `OTP` layers. Note that depending on the *voltage* decay $\tau_{\text{vol}}$, the CUBA neurons will function either as `LIF` or `IF` neurons.

$$U_i[t] = (1 - \tau_{\text{cur}})U_i[t-1] + \sum_j W_{i,j} S_j[t] \quad , \quad V_i[t] = (1 - \tau_{\text{vol}})V_i[t-1] + U_i[t] \quad \text{(10a)}$$

$$S_i[t] = V_i[t] \geq V_{\text{thr}} \quad , \quad V_i[t] = V_i[t](1 - S_i[t]) \quad \text{(10b)}$$

# D   Spike-train synchrony tests

Here we present the spike-train synchrony tests to validate our quantization of the LDN. Spike-train synchrony metrics measure the extent of *temporal alignment* of *two* spike-trains, i.e., how *similar* or *dissimilar* they are; for which, a number of methods exists [38], we pick *three*: (1) Victor-Purpura distance [39] (VPR-dist) – a popular time-scale dependent metric, (2) Inter-Spike Interval distance [40] (ISI-dist) – a time-scale independent metric, & (3) SPIKE-synchrony [41] (SPK-sync) – another time-scale independent metric (libraries: PySpike [42] & `Elephant` [43]) For any two spike-trains:

- VPR-distance metric measures the *dissimilarity* between them by computing the minimum cost of transforming a given spike-train into another – based on certain defined operations. For highly *dissimilar* spike-trains, VPR-distance metric will be high, and vice-versa.

- ISI-distance metric also measures the *dissimilarity* between them by using inter-spike intervals to estimate instantaneous local firing rates of spike-trains and quantifying that difference. For highly *dissimilar* spike-trains, ISI-distance will be high, and vice-versa.

- SPIKE-synchrony metric measures the *similarity* between them by using coincidence windows to determine if a pair of spikes, each from two different spike-trains are coincident or not. For highly *similar* spike-trains, SPIKE-synchrony will be high, and vice-versa.

We now explain the *procedure* of our spike-train synchrony tests. We consider spike-trains generated from *four* implementations of LDN: *Two* Python implementations – (1) Original Continuous-valued LDN implemented on CPU (OC-LDN-CPU-Py) & (2) Quantized LDN implemented on CPU (QT-LDN-CPU-Py), and *two* `Lava` implementations – (3): Quantized LDN implemented on CPU (QT-LDN-CPU-Lv) & (4): Quantized LDN implemented on Loihi-2 embedded LMT cores (QT-LDN-LMT-Lv). Note that in our experiments, we have utilized *three* (all quantized) implementations of LDN, i.e., QT-LDN-CPU-Py while training, and QT-LDN-CPU-Lv & QT-LDN-LMT-Lv while evaluation on Loihi-2's *simulation* & *physical* backends respectively. For comparisons among the *four* implementations of LDN, we consider all the 6 pairs (i.e., taking two at a time), denoted below:

- *Comparisons between Continuous-valued LDN and its Quantized implementations*

1. $OC_{\text{py}}^{\text{cpu}}$-vs-$QT_{\text{py}}^{\text{cpu}}$: Comparison between OC-LDN-CPU-Py and QT-LDN-CPU-Py
2. $OC_{\text{py}}^{\text{cpu}}$-vs-$QT_{\text{lv}}^{\text{cpu}}$: Comparison between OC-LDN-CPU-Py and QT-LDN-CPU-Lv
3. $OC_{\text{py}}^{\text{cpu}}$-vs-$QT_{\text{lv}}^{\text{lmt}}$: Comparison between OC-LDN-CPU-Py and QT-LDN-LMT-Lv

- *Comparison between two Quantized LDNs in Lava: one on CPU and another on LMT*

4. $QT_{\text{lv}}^{\text{cpu}}$-vs-$QT_{\text{lv}}^{\text{lmt}}$: Comparison between QT-LDN-CPU-Lv and QT-LDN-LMT-Lv

- *Comparisons between Quantized LDNs in Python and Lava (on CPU & LMT)*

5. $QT_{\text{py}}^{\text{cpu}}$-vs-$QT_{\text{lv}}^{\text{cpu}}$: Comparison between QT-LDN-CPU-Py and QT-LDN-CPU-Lv
6. $QT_{\text{py}}^{\text{cpu}}$-vs-$QT_{\text{lv}}^{\text{lmt}}$: Comparison between QT-LDN-CPU-Py and QT-LDN-LMT-Lv

With respect to designing the LDN, note that it is characterized by two values: '$d$' & '$\theta$'; for our tests, we consider all the values of $d$ & $\theta$ over which we tune our LSNN, i.e., $d \in \{4, 6, 8, 10, 12, 14, 16, 24\}$ and $\theta \in \{110, 130, 150\}$, thus, a total of $8 \times 3 = 24$ combinations or 24 LDNs. Consider one input signal and *two* different *implementations* of an LDN, we get *two sets* - each of $d$-dimensional state-vectors $\mathbf{x}[t]$ – that are encoded to binary spikes via the two-neuron encoder system (ENC in LSNN), thus producing *two sets* of $(2 \times d)$-dimensional spike-trains – i.e., one *set* for

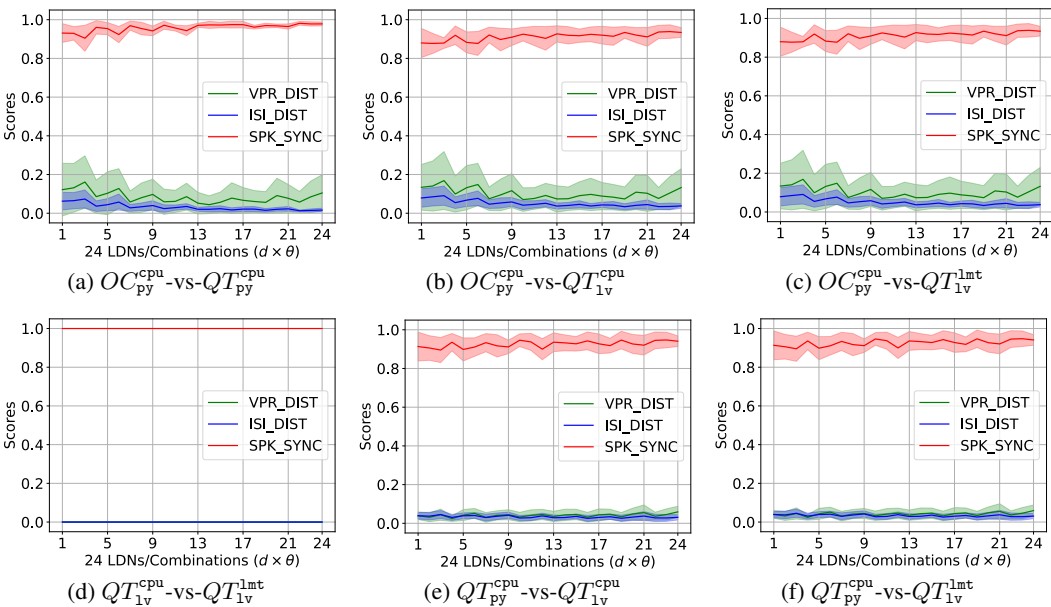

**Figure 4:** *Spike-train synchrony tests.* All 6 pair-wise comparisons of *four* LDN implementations. For VPR-dist & ISI-dist, lower is better, and for SPK-sync, higher is better. Solid line is mean and shaded region is std.

each *implementation* of LDN. Note that a scalar similarity/dissimilarity *score* is computed between *two* individual spike-trains. Therefore, we align those *two sets* of spike-trains dimension-wise, and $\forall$ dimensions $i \in (2 \times d)$ we compute a $score_i$ between the *two* $i^{\text{th}}$-dimension's spike-trains followed by the mean of all $score_i$, thus producing one *final-score* – representing how similar or dissimilar are the $(2 \times d)$ spike-trains on average (for the considered LDN and input signal). Likewise, for the same input signal we compute 24 similarity/dissimilarity *final-scores* for all the 24 LDNs. We repeat this process for 75 randomly chosen signals (5 each from our 15 datasets), thereby obtaining 75 such 24-dimensional *final-scores*; we then again compute mean and std across all these 75 vectors. Thus, for a considered comparison (out of the six above), this averaged 24-dimensional *final-scores* represents how similar/dissimilar are the two compared *implementations* of LDN over the 15 datasets. For all our 6 comparisons, Fig 4 below shows the (considered) 3 types of spike-train similarity/dissimilarity mean *final-scores* (and corresponding std) for all the 24 LDNs averaged over all 75 random signals.

As can be seen above, in Fig 4a, our quantized LDN implementation ($QT_{\text{py}}^{\text{cpu}}$) is quite similar to the continuous-valued LDN ($OC_{\text{py}}^{\text{cpu}}$), where the mean SPK-sync scores tend to be near 1 and the mean ISI-dist & VPR-dist scores are closer to 0. Coming to the Figs 4b & 4c, we see that both are same, this is because the quantized implementations of LDN via Lava on CPU ($QT_{\text{lv}}^{\text{cpu}}$) & LMT ($QT_{\text{lv}}^{\text{lmt}}$) are same; this is further proved in the Fig 4d where we see that ISI-dist & VPR-dist are exactly 0 and SPK-sync is exactly 1, i.e., perfect synchrony. However, note that the Figs 4b & 4c are *different* from Fig 4a – ideally, this should *not* be the case, as the underlying maths for $QT_{\text{py}}^{\text{cpu}}$, $QT_{\text{lv}}^{\text{cpu}}$, & $QT_{\text{lv}}^{\text{lmt}}$ is same; the *difference* exists because of our programming implementations in Python & Lava.

In our Python implementations ($OC_{\text{py}}^{\text{cpu}}$ & $QT_{\text{py}}^{\text{cpu}}$), in the *first* time-step (of the input signal's duration), the signal's *first* element is rightly accounted for LDN's state-vector generation and encoding to spikes, whereas, in our Lava implementations ($QT_{\text{lv}}^{\text{cpu}}$ & $QT_{\text{lv}}^{\text{lmt}}$), in the *first* time-step, we generate an *all zero* state-vector, thus (certainly) *no* spikes; rather, from the *second* time-step onwards the actual input signal is accounted for LDN's state-vector generation and encoding to spikes. Therefore, $QT_{\text{lv}}^{\text{cpu}}$ & $QT_{\text{lv}}^{\text{lmt}}$ run one time-step *late* than $OC_{\text{py}}^{\text{cpu}}$ & $QT_{\text{py}}^{\text{cpu}}$. This behavior is partly because of the unique `LoihiProtocol` that our Lava `Process` - for inputting signals to the LDN follows. In this `Process`, every time-step, the `run_spk()` phase sends a contiguous scalar of the input signal to the LDN. Note that, by the `LoihiProtocol`'s design, in every time-step, `run_spk()` phase is the *first* phase to be executed. Since in the first time-step, the actual signal is yet to be initialized and the `run_spk()` phase executes first, it sends a zero scalar to the LDN; the next phase to execute (in the same first time-step) in our implementation is `run_post_mgmt()`, where the input signal then gets initialized. Thus, from the *second* time-step onwards, the actual signal is accounted for state-vectors and subsequent spike generation. We opted for such Lava implementation for our experimental ease, however, one can choose to implement $QT_{\text{lv}}^{\text{cpu}}$ & $QT_{\text{lv}}^{\text{lmt}}$ to match $QT_{\text{py}}^{\text{cpu}}$ outputs. In the Figs 4e &

4f, we explicitly compare our $QT_{\text{py}}^{\text{cpu}}$ with $QT_{\text{lv}}^{\text{cpu}}$ & $QT_{\text{lv}}^{\text{lmt}}$, where, we see that both the plots are expectedly same, and the spike-trains from $QT_{\text{lv}}^{\text{cpu}}$ & $QT_{\text{lv}}^{\text{lmt}}$ are quite similar to those from $QT_{\text{py}}^{\text{cpu}}$.

Overall, Fig 4a shows the difference purely due to quantization, Figs 4b & 4c show the difference due to quantization and one time-step late processing, Fig 4d shows that $QT_{\text{lv}}^{\text{cpu}}$ & $QT_{\text{lv}}^{\text{lmt}}$ are exactly same, and finally, Figs 4e & 4f show the difference (only) due to one time-step late processing.

## E `ProcessModel`s of *LavaLSNN*

The `ProcessModels` of our defined *InpSigToLdn*, *LdnEncToSpk*, and *OutSpkToCls* `Processes` of *LavaLSNN* are briefly described below; their implementations are in our code [32].

- *InpSigToLdn*: For this `Process`, we implement only *one* `ProcessModel`: *PyInpSigToLdnModel* that is inherited from `lava`'s `PyLoihiProcessModel` and executes on CPU. It is responsible for feeding an input signal to the LDN (that runs either on CPU or on LMT).

- *LdnEncToSpk*: For this `Process` (considering *simulated* and *physical* Loihi-2), we implement *two* `ProcessModels`: (a) *CLdnEncToSpkOnLmtModel* that is inherited from `lava`'s `CLoihiProcessModel` and executes on LMT, and (b) *PyLdnEncToSpkOnCpuModel* that is inherited from `lava`'s `PyLoihiProcessModel` and executes on CPU. Depending on the backend, only *one* of the two `ProcessModels` is executed. It is responsible for computing the quantized state-vectors $\bar{\mathbf{x}}[t]$ i.e., temporal features and encoding these to binary spikes.

- *OutSpkToCls*: For this `Process`, we implemented only one `ProcessModel`: *PyOutSpkToClsModel* that is inherited from `lava`'s `PyLoihiProcessModel` and executes on CPU. It is responsible for collecting the output spikes from *NetxSNN* and inferring classes/accuracy.

When *LavaLSNN* is deployed on Loihi-2's *simulation* i.e., on CPU (via `Loihi2SimCfg`), following `ProcessModels` (of the constituent `Processes`) are executed: *PyInpSigToLdnModel*, *PyLdnEnc-ToSpkOnCpuModel*, and *PyOutSpkToClsModel*. And when *LavaLSNN* is deployed on the *physical* Loihi-2 chips (via `Loihi2HwCfg`), following `ProcessModels` (of the constituent `Processes`) are executed: *PyInpSigToLdnModel*, *CLdnEncToSpkOnLmtModel*, and *PyOutSpkToClsModel*. All these `ProcessModels` follow the `LoihiProtocol` for synchronization. Note that we *have to* deploy the signal-input `Process`: *InpSigToLdn* and output-spikes collection `Process`: *OutSpkToCls* (in case of both - Loihi-2's *simulation* and *physical* hardware) on CPU – as these two `Processes` are the I/Os. Also note that Lava automatically deploys `Processes` obtained from `lava-dl`'s `netx` API (in our case - the *NetxSNN* `Process` obtained from `slayer`-trained *SlayerSNN*) on the appropriate backends depending on the choice of: `Loihi2SimCfg` (on CPU) & `Loihi2HwCfg` (on Neuro-Core).

## F *CLdnEncToSpkOnLmtModel* for deploying `LDN` and `ENC` layers on LMT

Documentation for programming Loihi-2 chips via Lava can be found at `https://lava-nc.org/`. On one hand, where the documentation to program Neuro-Cores is well presented there, on the other, *little to no* documentation to program LMT cores is provided – this paucity limits the researchers to use LMT cores for their Loihi-2 deployments, and prompts them to otherwise use the CPU for implementing *non-spiking* operations or *encoding* continuous values to spikes. Note that the embedded LMT cores are *more* power-efficient than CPUs. In Algorithm 1, we present our *CLdnEncToSpkOnLmtModel* that implements a quantized SSM (i.e., LDN) on LMT cores. Our implementation also serves as an *addition* to the limited documentation on programming LMT cores.

We next explain our code in Algorithm 1 (Alg-1, note: only 32-bit ops are supported on LMT) by first recalling that the `CLoihiProcessModel`: *CLdnEncToSpkOnLmtModel* has a corresponding `Process`: *LdnEncToSpk* that provides the interface to communicate with other `Processes`. Note that the C file (in our case: `ldn_enc_to_spk_on_lmt.c`) containing the code in Alg-1 *must* have a corresponding *header* file bearing the same file name (in our case: `ldn_enc_to_spk_on_lmt.h`), see line 1 in Alg-1. The *header* file should contain the function prototypes of all `LoihiProtocol` phases used in the C file. One should also include another *header* file with name `predefs_X.h`, where X is the name of `CLoihiProcessModel` of the `Process` to be executed on LMT; in our case X is *CLdnEncToSpkOnLmtModel* (see line 2). Note that the file `predefs_CLdnEncToSpkOnLmtModel.h` is auto-generated containing the variables of the corresponding `Process` (in our case: *LdnEncToSpk*).

As can be seen in Alg-1 (lines 8,12,27,&32), one can implement each phase of the `LoihiProtocol` (on LMT) with an argument `runState *rs` passed to it. The current time-step of the execution can be accessed via the `time-step` attribute of `*rs`, which can be used to implement time-step dependent ops (lines 13, 34). One can also define global variables (lines 5, 6) and custom functions

(line 19) in the C file, as well as, access the variables defined in the corresponding Lava Process by their same name (lines 13, 20, i.e., ps_ts & ORDER) – this is because of the inclusion of auto-generated header file predefs_CLdnEncToSpkOnLmtModel.h. Note that all the functions can access all global variables and Process variables. An important thing to consider is that the Process running on LMT cores communicates with Processes running on CPUs and Neuro-Cores. In our case, the LMT Process receives a quantized input (i.e., the input signal $\bar{\mathbf{u}}[t]$) from an input Process on CPU; for the same, the recv_vec_dense() function (line 40) can be used that accepts the *rs and two other variables: sig_inp & input. Note that sig_inp is an input port (defined in *LdnEncToSpk* Process) that receives a signal from the CPU Process, and input is a local variable that stores the received signal for local processing. Subsequently, when the input (for the current time-step) is processed and spikes are generated, they can be sent from the LMT Process to the Neuro-Core Process via the send_vec_dense() function (line 83) that accepts *rs and two other variables: spk_out & spike_data. Note that spk_out is an output port (defined in *LdnEncToSpk* Process) that sends the spikes stored in local variable spike_data to the Neuro-Core Process.

Algorithm 1: C code of our *CLdnEncToSpkOnLmtModel* to implement LDN (and ENC) on LMT

```c
1   #include "ldn_enc_to_spk_on_lmt.h"
2   #include "predefs_CLdnEncToSpkOnLmtModel.h"
3
4   // Define the global LDN state vector with maximum ORDER = 128.
5   int32_t g_ldn_state[128] = {0};
6   int32_t g_volt[256] = {0}; // Set twice the ORDER number of spiking neurons.
7
8   int spk_guard(runState *rs) {
9     return 1; // keep running the `run_spk()` every time-step.
10  }
11
12  int post_guard(runState *rs) {
13    if (rs->time_step % ps_ts[0] == 1) // Presentation time of one sample is over.
14      return 1;
15
16    return 0;
17  }
18
19  void zero_out_global_arrays() {
20    for(uint32_t i=0; i<ORDER[0]; i++)
21      g_ldn_state[i] = 0;
22
23    for(uint32_t i=0; i<2*ORDER[0]; i++)
24      g_volt[i] = 0;
25  }
26
27  void run_post_mgmt(runState *rs) {
28    zero_out_global_arrays();
29  }
30
31  // Following function is called every time-step.
32  void run_spk(runState *rs) {
33
34    if (rs->time_step % ps_ts[0] == 1) // Presentation time of one sample is over.
35      zero_out_global_arrays();
36
37    int32_t input[sig_inp.size];
38    uint32_t spike_data[spk_out.size];
39
40    recv_vec_dense(rs, &sig_inp, input); // Get u[t].
41
42
```

```
43    // Note that the matrices Ap, Bp, encoders E, and ORDER are already defined
44    // in Process. The variable ORDER can be accessed as *ORDER or ORDER[0].
45
46    // Compute Ap * x[t] and Bp * u[t].
47    int32_t Apx[*ORDER], Bpu[*ORDER];
48    for(uint32_t i=0; i< *ORDER; i++) {
49      int32_t sum = 0;
50      for(uint32_t j=0; j< *ORDER; j++) {
51        sum += (Ap[i][j] * g_ldn_state[j]);
52      }
53      Apx[i] = sum;
54      Bpu[i] = Bp[i][0] * input[0];
55    }
56
57
58    // Compute Apx[t] + Bpu[t].
59    for (uint32_t i=0; i< *ORDER; i++) {
60      int32_t state = Apx[i] + Bpu[i];
61      g_ldn_state[i] = (
62              //Ceil Integer Division
63              (state > 0) ? (1 + (state-1)/ *scale_factor) : (state/ *scale_factor)
64          );
65    }
66
67    // Rate Encode the current time-step's ldn_state to the spikes. Note that the
68    // g_gain, g_bias, and g_v_thr are already defined in the the Process.
69    for(uint32_t i=0; i< 2* *ORDER; i++) {
70      int32_t J = *g_gain * E[i] * g_ldn_state[i/2] + *g_bias; // g_bias = 0 here.
71      g_volt[i] = g_volt[i] + J; // Update the global voltage state of IF neurons.
72      if(g_volt[i] > *g_v_thr) {
73        spike_data[i] = 1;
74        g_volt[i] = 0;
75      }
76      else {
77        spike_data[i] = 0;
78        if(g_volt[i] < 0)
79              g_volt[i] = 0; // Rectifiy the voltage.
80      }
81    }
82
83    send_vec_dense(rs, &spk_out, spike_data);
84  }
```

# G  Datasets

We have conducted our experiments on 15 univariate TSC datasets – publicly available at `https://www.timeseriesclassification.com/dataset.php`. Note that our chosen datasets fall into *five* categories of applications: *ECG*, *Sensor*, *Spectrograph*, *Motion*, & *Device*. On average, these datasets have close to zero mean and unit variance [33]. Table 3 provides their brief description:

Table 3: All 15 datasets' tabular summary

| Acronym | Dataset | Signal Duration | Number of Classes | Train Size | Test Size | Category |
|---------|---------|-----------------|-------------------|------------|-----------|----------|
| E5000 | ECG5000 | 140 | 2 | 500 | 4500 | ECG |
| FordA | FordA | 500 | 2 | 3601 | 1320 | Sensor |
| FordB | FordB | 500 | 2 | 3636 | 810 | Sensor |
| Wafer | Wafer | 152 | 2 | 1000 | 6164 | Sensor |
| Eqaks | Earthquakes | 512 | 2 | 322 | 139 | Sensor |
| Cofe | Coffee | 286 | 2 | 28 | 28 | Spectrograph |
| ToSe1 | ToeSegmentation1 | 277 | 2 | 40 | 228 | Motion |
| ToSe2 | ToeSegmentation2 | 343 | 2 | 36 | 130 | Motion |
| Light2 | Lightning2 | 637 | 2 | 60 | 61 | Sensor |
| Wrms | Worms | 900 | 5 | 181 | 77 | Motion |
| WrTC | WormsTwoClass | 900 | 2 | 181 | 77 | Motion |
| OlOil | OliveOil | 570 | 4 | 30 | 30 | Spectrograph |
| Meat | Meat | 448 | 3 | 60 | 60 | Spectrograph |
| Comp | Computers | 720 | 2 | 250 | 250 | Device |
| RDev | RefrigerationDevices | 720 | 3 | 375 | 375 | Device |

Note: ECG5000 has 5 classes originally, one corresponding to *healthy* heartbeats and rest 4 falling into *unhealthy* heartbeats – we grouped these 4 into one *unhealthy* class, thus binarizing ECG5000 [36, 21].

# H  Intrusive experiments with LSNN

We conducted *two* intrusive experiments over LSNN: (a) with neuron's decays: $\tau_{cur}$ & $\tau_{vol}$ *too* set to trainable (along with the weights), and (b) using *continuous* valued LDN in LSNN (while training and evaluation on GPU); results for both are presented below in Tabs 4 & 5 (respectively). After comparing these results with those in Tab 1 above, we observed that training neuron decays (along with the weights) in LSNN is *not* decisively superior than training only weights, and training LSNN with continuous-valued LDN too, does *not* conclusively establish its superiority over training with quantized LDN (more experiments needed). Although, for some datasets, added trainable decays do appear to offer better *l2sim_best* accuracies; notably, for *ToSe2*, *Wrms*, *WrTC*, and *Meat* datasets, Tab 4 reports 95.39, 81.82, 88.31, 98.33 (%) compared to 93.85, 79.22, 85.71, 96.67 (%) in Tab 1. Therefore, we inferred on *physical* Loihi-2 with *ToSe2* dataset's *l2sim_best* model, and found that it achieves 93.08% (exactly as LSTM-FCN); this hints better *l2hw_best* results for other datasets too.

Table 4: Our results with LSNN on GPU and on Loihi-2 simulation with trainable decays & weights

| Method/ Accuracy | Datasets | | | | | | | | | | | | | | |
|---|---|---|---|---|---|---|---|---|---|---|---|---|---|---|---|
| | E5000 | FordA | FordB | Wafer | Eqaks | Cofe | ToSe1 | ToSe2 | Light2 | Wrms | WrTC | OlOil | Meat | Comp | RDev |
| *slyr_best* | 95.69 | 91.14 | 80.37 | 98.04 | 79.14 | 100.0 | 95.18 | 94.62 | 78.69 | 81.82 | 89.61 | 63.33 | 100.0 | 72.00 | 54.67 |
| *slyr_mean* | 95.08 | 90.82 | 78.37 | 97.65 | 76.12 | 96.43 | 91.40 | 88.31 | 70.82 | 75.84 | 84.68 | 46.68 | 96.00 | 67.84 | 49.39 |
| *l2sim_best* | 95.60 | 90.91 | 78.89 | 98.35 | 78.42 | 100.0 | 93.86 | 95.39 | 77.05 | 81.82 | 88.31 | 63.33 | 98.33 | 70.40 | 52.53 |
| *l2sim_mean* | 95.01 | 90.70 | 77.51 | 97.79 | 75.40 | 92.86 | 90.97 | 88.92 | 68.53 | 75.58 | 84.16 | 44.00 | 88.00 | 67.76 | 49.92 |

Table 5: Our results with LSNN on GPU where the LSNN comprises original *continuous*-valued LDN, and LSNN on Loihi-2 simulation where the LSNN comprises *quantized* LDN

| Method/ Accuracy | Datasets | | | | |
|---|---|---|---|---|---|
| | E5000 | FordA | FordB | Wafer | Eqaks |
| *slyr_best* | 95.71 | 91.36 | 79.63 | 98.07 | 79.14 |
| *slyr_mean* | 94.90 | 90.97 | 78.84 | 97.76 | 75.54 |
| *l2sim_best* | 95.80 | 91.06 | 77.90 | 98.20 | 77.70 |
| *l2sim_mean* | 95.03 | 90.61 | 77.51 | 97.95 | 75.25 |

# I  Procedure and distribution plots of LSNN's profiling on *physical* Loihi-2

Here, we explain our procedure to *energy-* and *latency-*profile the LSNN on *physical* Loihi-2 chip, as well as, present the frequency distribution plots of the profiled metrics. With respect to *LavaLSNN*'s (and LDN's) variations to profile, we fixed $\tau_{\mathrm{cur}}=\tau_{\mathrm{vol}}=0$ & $\theta=130$, and varied the order $d\in\{8, 16, 24\}$ - as $d$ determines the size of *LavaLSNN* (& LDN). $\forall\, d$, we consider all the combinations of $\mathcal{S}_j\in\{256, 512, 1024, 2048\}$ & $\mathcal{B}_k\in\{2, 3, 4\}$; and $\forall\,(\mathcal{S}_j, \mathcal{B}_k)$, we *run* the considered *LavaLSNN*'s components for 2000 *duplicates* of the hard-coded signal (of duration 140 time-steps (t.s.)) i.e., $28e4$ t.s. in total, and record the total energy consumed and execution time *on* Loihi-2 – each *run* is repeated 15 times (in real-time, each *run* executed for $30s$-$120s$). To obtain the *per-sample* energy & latency metrics of processing a 140 t.s. long signal on Loihi-2, we divide the total energy & execution time (of each *run*) by the number of *duplicates* i.e., 2000. Finally, $\forall\, d$, we report the (mean, std) of *per-sample* energy & latency metrics over all the 15 repeats of all 12 combinations $(\mathcal{S}_j, \mathcal{B}_k)$ in Tab 2.

We next present the frequency distribution plots of the LSNN's different components' energy & latency profiling - on different backends of *physical* Loihi-2, namely (1) LMT & Neuro-Core (combined), (2) only LMT, and (3) only Neuro-Core. Note that, the LDN & ENC layers, both execute on LMT, where the LDN has $\mathcal{O}(d^2)$ operations (matrix multiplied to a vector – lines 46-55 in Algorithm 1), and the ENC has $\mathcal{O}(d)$ operations (lines 67-81 in Algorithm 1). Therefore, the LDN accounts for (or contributes to) much of the energy & latency metrics values. The $\mathcal{O}(d^2)$ also explains why the metric values in Tab 2 (for (1) & (2)) for $d = 8$ & 16 are similar, and clearly different for $d$=24. On a side note, we conducted profiling experiments with $\mathcal{B}_k = 1$ time-step binning too, however, the results were either not consistent or the simulation threw runtime errors for lower buffer size $\mathcal{S}_j$.

## I.1  Energy distribution plots for (1), (2), & (3) in Tab 2

### I.1.1  For (1) LMT & Neuro-Core: By *LdnEncToSpk→NetxSNN*

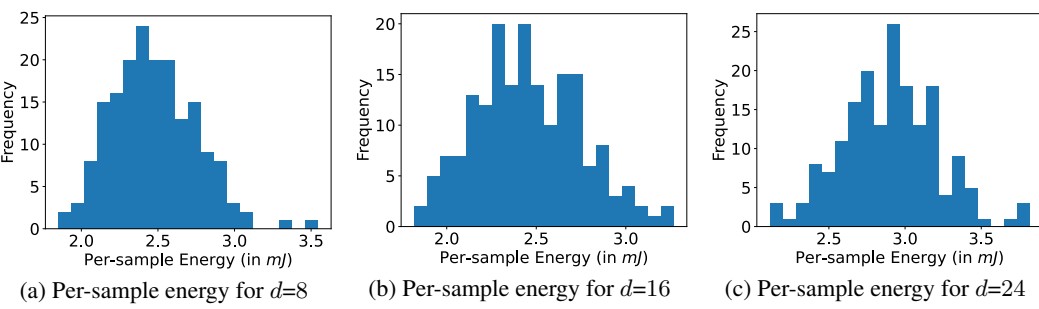

(a) Per-sample energy for $d$=8    (b) Per-sample energy for $d$=16    (c) Per-sample energy for $d$=24

Figure 5: Frequency distribution of per-sample energy consumed on LMT & Neuro-Core

### I.1.2  For (2) only LMT: By *LdnEncToSpk*

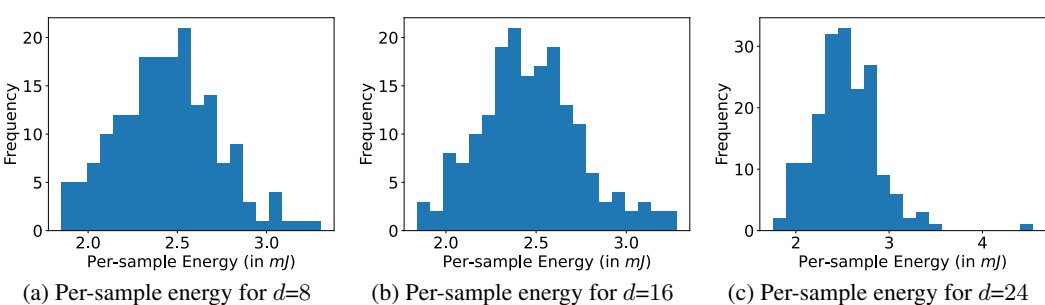

(a) Per-sample energy for $d$=8    (b) Per-sample energy for $d$=16    (c) Per-sample energy for $d$=24

Figure 6: Frequency distribution of per-sample energy consumed on only LMT

### I.1.3 For ⟨3⟩ only Neuro-Core: By *NetxSNN*

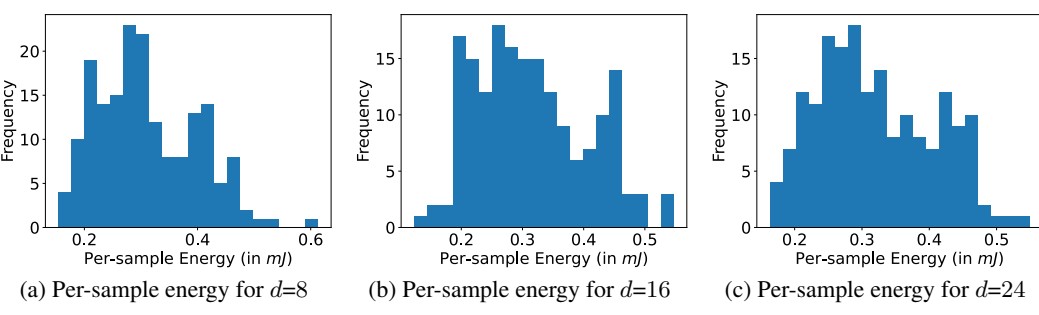

(a) Per-sample energy for *d*=8      (b) Per-sample energy for *d*=16      (c) Per-sample energy for *d*=24

Figure 7: Frequency distribution of per-sample energy consumed on only Neuro-Core

## I.2 Latency distribution plots for ⟨1⟩, ⟨2⟩, & ⟨3⟩ in Tab 2

### I.2.1 For ⟨1⟩ LMT & Neuro-Core: By *LdnEncToSpk*→*NetxSNN*

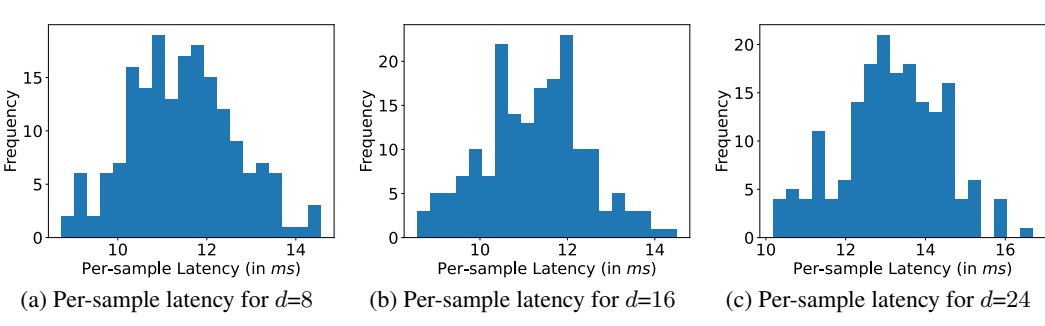

(a) Per-sample latency for *d*=8      (b) Per-sample latency for *d*=16      (c) Per-sample latency for *d*=24

Figure 8: Frequency distribution of per-sample processing latency on LMT & Neuro-Core

### I.2.2 For ⟨2⟩ only LMT: By *LdnEncToSpk*

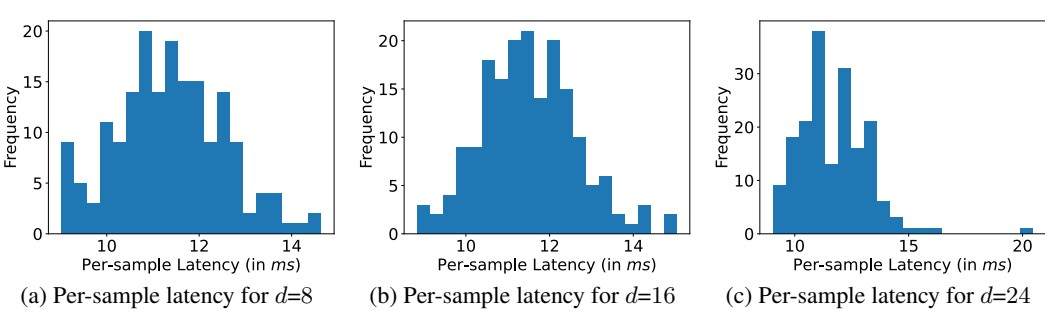

(a) Per-sample latency for *d*=8      (b) Per-sample latency for *d*=16      (c) Per-sample latency for *d*=24

Figure 9: Frequency distribution of per-sample processing latency on only LMT

### I.2.3 For ⟨3⟩ only Neuro-Core: By *NetxSNN*

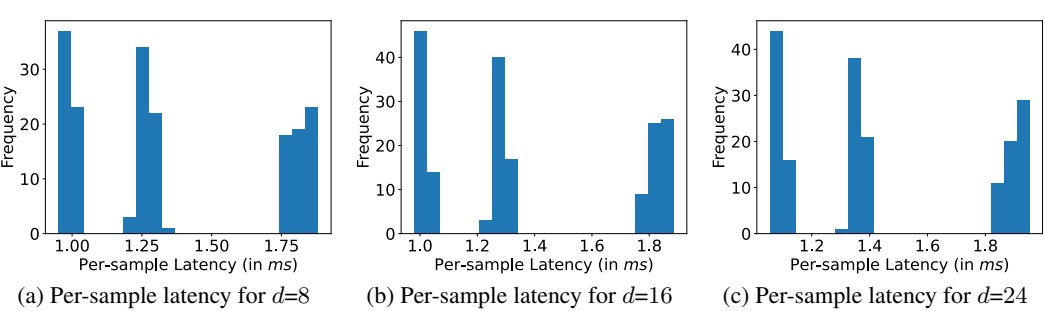

(a) Per-sample latency for *d*=8      (b) Per-sample latency for *d*=16      (c) Per-sample latency for *d*=24

Figure 10: Frequency distribution of per-sample processing latency on only Neuro-Core

# J  Profiling of a recurrent spiking network – representative of LDN

Here, we present and profile a recurrent spiking network – *architecturally-representative* of LDN, on Loihi-2 Neuro-Cores. That is, the spiking network in Fig 11 below is *not* an actual implementation of LDN, rather, a representation of how the architecture of *spiking* LDN could look like *if* implemented via NEF – picked up from [36]. The rectangle and circle in Fig 11 are *Ensembles* of "Leaky Integrate & Fire" (LIF) neurons that represent the univariate signal $\bar{\mathbf{u}}[t]$ and the $d$-dimensional state vector $\bar{\mathbf{x}}[t]$ respectively, – via their spiking activity (NEF Principle 1). The outputs from the *Ensembles* are linearly transformed (NEF Principle 2) by the representative random matrices $\mathbf{A^r}$ & $\mathbf{B^r}$ (of $\bar{\mathbf{A}}$ & $\bar{\mathbf{B}}$ respectively in Eq (5)). In our simulation, the number of neurons in the *Ensembles* for $\bar{\mathbf{u}}[t]$ and $\bar{\mathbf{x}}[t]$ depend on $d$, i.e., $2{\times}d$ and $25{\times}d$ respectively. Upon profiling such a recurrent spiking network on Loihi-2 Neuro-Cores, we found that its energy (& latency) metrics (see Tab 6) were similar to that of (3) in Tab 2. This suggests that if such a recurrent spiking network implements the LDN in LSNN, then the overall energy-efficiency of the *entire* LSNN on Loihi-2's Neuro-Cores would significantly be improved. The per-sample energy metric distribution of the recurrent network is in Sec J.1.

Table 6: Per-sample Energy metrics of the recurrent spiking network in Fig 11

|  | only Neuro-Core | | |
|---|---|---|---|
| Order $d\rightarrow$ | 8 | 16 | 24 |
| Energy $(mJ)$ | 0.30, 0.08 | 0.33, 0.09 | 0.32, 0.09 |

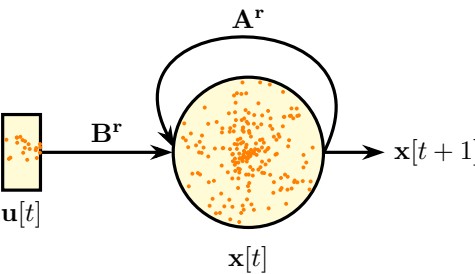

Figure 11: NEF-based representational spiking-architecture of LDN

## J.1  Frequency distribution of energy metric of network in Fig 11 - on *physical* Loihi-2

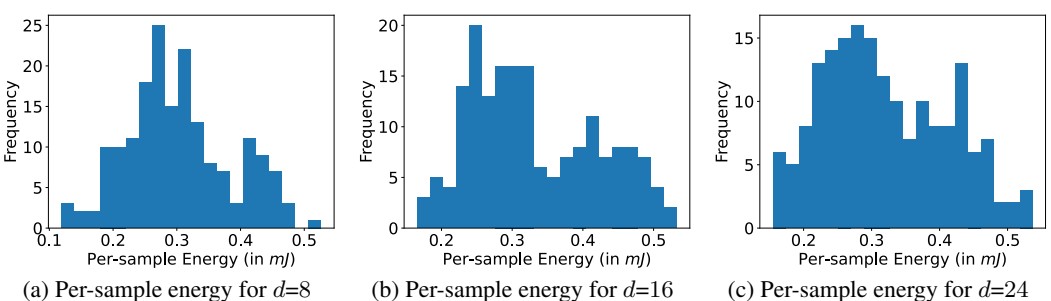

(a) Per-sample energy for $d=8$  (b) Per-sample energy for $d=16$  (c) Per-sample energy for $d=24$

Figure 12: Frequency distribution of per-sample energy consumed on only Neuro-Core

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
