# OpenReview forum: "Legendre-SNN on Loihi-2: Evaluation and Insights"
_NeurIPS.cc/2024/Workshop/MLNCP — MLNCP Poster_

### Official Review · Reviewer_JSjr · 2024-10-04
**This work presents the implementation of Legendre Spiking Neural Networks (LSNN) on the Loihi 2 neuromorphic system for time series classification using both Neuro-Cores and the Lakemount CPUs.**

**Rating:** 5
**Confidence:** 5

**Review:**

This work presents the implementation of Legendre Spiking Neural Networks (LSNN) on the Loihi 2 neuromorphic system for time series classification using both Neuro-Cores and the Lakemount CPUs. The authors describe how they modify the previously proposed LSNN architecture to match the capabilities of Loihi 2. The implementation of the Legendre Delay Network (LDN) on the Lakemount cores, which only offer 32-bit integer operations, is described in more detailed, including how to tune the scaling factors of this state-space model during quantization. 15 different time series classification are evaluated regarding model accuracy and also regarding energy and latency on Loihi 2. The methodology of the implementation is completely described, but hard to follow due to many acronyms. The contribution of the different cores to latency and energy of the full applications are studied. Instead, a comparison to the energy consumption on other systems is missing.

# quality
Overall, the authors present the concept and a lot of engineering work of the implementation of the LSNN on Loihi 2. They study 15 different benchmarks and mainly provide analysis of the model accuracies of different variants of the model (Software simulation of LSNN vs. hardware simulation of Loihi vs. Loihi 2 implementation). In addition to providing the full pipeline for the deployment, the main contribution is to showcase how the Lakemount cores of Loihi 2 can be leveraged for hybrid compute loads consisting of spiking and non-spiking components. One insight is that using Lakemount cores increases the power consumption in comparison with the NeuroCores.
It is unclear why the accuracy is mainly compared to the LSTM-FCN work, but not to other work. The authors should add their reasons behind that. The comparison to other work (l234-238) is very brief and lacks clarity.
A energy and latency comparison of the Loihi2 solution with other edge systems is completely missing. The authors could at least try to compare with the results from [Gaurav et al 2023]. Such comparison would help to show the benefits of the proposed solution.

# clarity
Overall, the paper is very hard to read due to abundant use of acronyms and abbreviations. The paper is cluttered with too many results / details considering the 6-page limit.

# originality
The authors build upon prior work [Gaurav et al 2023] who introduced the LSNN. The authors suggest modification for the Loihi 2 implementation. Especially they propose a strategy for implementing the LDN part using integer arithmetics in the x86 cores on Loihi 2.

# significance
The authors describe how to integrate the Lakemount core for hybrid applications on Loihi 2. This will be helpful for other users of Loihi 2.
As the energy and latency results are not compared to other hardware platforms, it is unclear how much faster or how much more energy-efficient the Loihi2 implementation is compared to other systems.

# pros
- LSNN implementation on Loihi 2 for time-series classification
- Use of Lakemount cores in addition to Neuro-Cores
- Very detailed assessment of different variants of own work, including accuracy, energy and latency
- code has been shared

# cons
- paper lacks clarity
- significance unclear as comparison to state-of-the-art is incomplete

# detailed feedback:
- Introduction: the 2nd bullet point of the contributions needs to be checked for grammar/language
- section 2.3 on LDN is very dense, makes it hard to read
- throughout the paper: many capitalized words that rather should be written in lower-case
- Table 1: prefixes “slyr”, “l2sim” “l2h2” and “LSTM-FCN” should be introduced in the table caption.
- Why comparison to LSTM-FCN only and not s.th. more recent?
- 4.3 Results analysis: It is unclear following which rule the results in table were highlighted in bold font. Why is there no highlight for E5000?
- 5 Discussion, introduction: it is strange that the authors point to the intrusive experiments. It would be better to add these experiments to Section 4.3.
- 5.1 Profiling LSNN on physical Loihi-2 chip:
  - Profiling on Loihi-2 is described in very detail, which is hard to follow, e.g., regarding the buffer size and binning B. I suggest moving the measurement details to the appendix and focus on the energy & latency results in the main paper. In my opinion, these results should rather be part of section 4, and not the discussion.
  - It would be great to know how many Neuro-Cores and LMT cores were used for the experiments in Table 2
- Overall Feedback: For a 6-page conference paper I recommend to reduce the amount of technical details and maybe even the number of benchmarks. Instead, use the gained space for better explaining the research.

---

### Decision · Program_Chairs · 2024-10-10

Accept (Poster)